# ZERO-SHOT POLICY TRANSFER WITH DISENTANGLED ATTENTION

## ABSTRACT

Domain adaptation is an open problem in deep reinforcement learning (RL). Often, agents are asked to perform in environments where data is difficult to obtain. In such settings, agents are trained in similar environments, such as simulators, and are then transferred to the original environment. The gap between visual observations of the source and target environments often causes the agent to fail in the target environment. We present a new RL agent, SADALA (Soft Attention DisentAngled representation Learning Agent). SADALA first learns a compressed state representation. It then jointly learns to ignore distracting features and solve the task presented. SADALA's separation of important and unimportant visual features leads to robust domain transfer. SADALA outperforms both prior disentangled-representation based RL and domain randomization approaches across RL environments (Visual Cartpole and DeepMind Lab).

## 1 INTRODUCTION

RL agents learn to maximize rewards within a task by taking actions based on observations. The advent of deep learning has enabled RL agents to learn in high dimensional feature spaces and complex action spaces (Krizhevsky et al., 2012; Mnih et al., 2016). Deep RL methods have beat human performance in a variety of tasks, such as Go (Silver & Hassabis, 2016). However, deep RL has two crippling drawbacks: high sample complexity and specificity to training task.

There are many domains where data collection is expensive and time consuming, such as healthcare, autonomous vehicles, and robotics (Gottesman et al., 2019; Barrett et al., 2010). Thus, agents are often trained in simulation and must transfer the resulting knowledge to reality. While this solves the issue of sample complexity, reality and simulated domains are sufficiently different that it is infeasible to naively train a deep RL agent in a simulation and transfer. This is known as the reality gap (Sadeghi & Levine, 2016). Jumping the reality gap is difficult for two orthogonal reasons. The first is that dynamics of a simulation are an approximation to the dynamics of the real world. Prior work has shown success in transfer between domains with different dynamics (Killian et al., 2017; Yao et al., 2018; Doshi-Velez & Konidaris, 2016). In this paper, we address the second difficulty: the difference in visual observations of states. Due to limitations in current photorealistic rendering, simulation and the real world are effectively two different visual domains (Sadeghi & Levine, 2016).

We present a method of robust transfer between visual RL domains, using attention and a $\beta$ variational autoencoder to automatically learn a state representation sufficient to solve both source and target domains. By learning disetangled and relevant state representation, our approach does not require target domain samples when training. The state representation enables the RL agent to attend to only the relevant state information and ignore all other, potentially distracting information.

## 2 RELATED WORK

Domain randomization is currently the most popular transfer method between different visual domains in RL (Tobin et al., 2017; OpenAI et al., 2018; Sadeghi & Levine, 2016). By training on many source domains, an RL agent implicitly learns to ignore factors of variation present. Sample complexity scales with the number of source domains and randomized factors. Modes of variation are maually selected. When faced with different variation, domain randomization will fail.

Furthermore, there is evidence that domain randomization can destablize the training of some RL algorithms, such as A3C and DDPG (Matas et al., 2018; Zhang et al., 2019). Though domain randomization has shown success in transfer, it has three downsides: high sample complexity, manual selection of irrelevant features, and ustable training (Tobin et al., 2017; OpenAI et al., 2018).

Some visual domain adaptation work is based on image to image translation (Ajakan et al., 2014; Tzeng et al., 2017; Hoffman et al., 2017; Zhu et al., 2017; Pan et al., 2017; Isola et al., 2017; Liu et al., 2017). These methods learn a mapping from source domain inputs to target domain inputs, utilizing adversarial methods, such as Generative adversarial Networks trained on data from both the source and target domain. During inference, the target domain input is translated to the source domain before being processed. While these methods have shown promising results, they have two downsides. First, translation incurs additional overhead at inference time. This overhead is especially large since the inputs are high-dimensional images. Second, these methods require samples from the target domain. In applications such as robotics, sampling the target domain is expensive and difficult to obtain. Further, the target domain may not be known at training time.

Some recent work focuses on modifying the inputs from the source domain so that they are similar to the target domain, effectively matching their distributions (Shrivastava et al., 2017). This approach uses a GAN to refine images from a (simulated) source domain to match images from the target domain: the real world. While this directly addresses domain shift and does not add any additional overhead at inference time in the target domain, it still assumes that samples from the target domain are present during training. It enables transfer between two differing visual domains, but it does not enable adaptation to previously unseen domains.

There is a body of work that takes this idea further. Rather than translating from the target domain to the source domain, it learns to map all domains to a canonical domain in which the RL agent operates (James et al., 2019). By learning to translate from randomized domains to a canonical domain, effectively training a translation network using domain randomization. While this approach has shown simulation to reality transfer, it icurrs overhead through the use of the mapping network; Rather than learning to map to a canoical domain, it would be more efficient to map to a disentagled representation of the canoical environment.

Other work learns to map image inputs to a latent space for use in RL. Specifically, it utilizes a $\beta$ Variational Autoencoder ($\beta$-VAE) to learn a compressed, disentagled state representation (Higgins et al., 2017). While this work focuses on domain adaptation in the RL setting, it utilizes a general-purpose generative model. Thus, the state representation is not tailored to the task posed; it preserves information needed to reconstruct the image that may not be needed to solve the RL task given. Additionally, unlike image translation, this method does not attempt to match the distribution of inputs from source and target domains. Since the $\beta$-VAE is trained to reconstruct input states, the distributions of compressed and input states will match. Accordingly, the distribution shift between source and target domains will be preserved, even when using compressed states. To circumvent this issue, the RL agent is trained on multiple source domains, as in domain randomization.

## 3 BACKGROUND

We formalize the concept of transfer between related MDPs. We denote the source domain as $D_S$ and the target domain as $D_T$. Both domains are MDPs, defined as $D = (S, A, T, R)$, where $S$ are states, $A$ are actions, $T$ is the transition function, and $R$ is the reward function. We assume that both $D_S$ and $D_T$ share a discount factor $\gamma$. In general transfer, states and actions may be quite different, while transition and reward functions are similar. We denote a family of related MDPs as $M$. Within $M$, transitions, rewards, and actions are the same across MDPs, and states are different. These states are generated from a set of latent state factors $S^z$, each passed through a different MDP-specific renderer $G_U$ for some MDP $U \in M$. In the case of visual cartpole, $S^z$ contains the position and velocity of the cart, the angle and position of the pole, and the colors of the cart, pole, axle, and track. The observed states are then rendered by the renderer. Specifically, $S_U = G_U(S^z)$ and $S_V = G_V(S^z)$, where $G_U$ and $G_V$ are any two MDPs within $M$.

In this paper, we target transfers where $T_S \approx T_T$, $R_S = R_T$, $A_S = A_T$, but $S_S = G_S(S^z) \neq S_T = G_T(S^z)$. Namely, when target dynamics are similar to source dynamics, rewards, actions, and true states are identical, but observed states $S_S$ and $S_T$ differ due to differing renderers $G_S$ and $G_T$.

# 4 SADALA

## 4.1 FEATURE EXTRACTION WITH $\beta$-VARIATIONAL AUTOENCODERS

Deep RL agents learn a mapping from states $s \in S$ to actions $a \in A$, either by directly maximizing reward or through a $Q$ function, using a deep neural network. The deep network implicitly learns a mapping function $F : S_U \to \hat{S}^z$, for some MDP $U \in M$ and a policy function $\pi : \hat{S}^z \to A$ that maps from latent state factors to actions. These functions are learned as a result of the feed-forward nature of deep neural networks. The policy $\pi$ is a function of the last layer of the network. The output of the last layer of the neural network is a highly processed feature representation of the input that is optimized for use in generating $\pi$. While this implicit mapping is sufficient for a deep RL agent to solve a single MDP from $M$, it is not general enough to transfer to other MDPs within $M$. By optimizing for $\pi$ $F$ overfits to the specific $S_U$ from the source domain $U$.

To circumvent this overfitting, DARLA proposed learning a feature extractor $F'$ that maps from any observed state $S_X$ to the latent state factors $S^z$ (Higgins et al., 2017). The RL agent is the trained on the (approximate) extracted latent state factors $\hat{S}^z$, separating the "vision" and "action" of the RL agent. DARLA uses a $\beta$ variational autoencoder to learn the mapping $F'$ (Higgins et al., 2017).

The extracted latent state factors $\hat{S}^z$ are (mostly) disentangled in practice, leading to better transfer results than a baseline RL agent. The factors each correlate with an independent factor in the true state $S^z$ of the task. However, it preserves all latent state variables $z \in S^z$, incorrectly assuming that they are all relevant to solving the MDP. When attempting to locate an object such as an enemy in a video game, an object for a robot, or a road for an autonomous vehicle, the position and orientation of the object are the most important factors. Visual distractions such as the color of the enemy, opacity of the object, or texture of the road are entirely irrelevant to its localization. Dependence on these distracting factors causes the RL agent to fail to robustly transfer to target visual domains.

## 4.2 ADDING AN ATTENTION MECHANISM

SADALA is similar to DARLA, but has one key difference—it explicitly learns to ignore irrelevant extracted latent state variables.

SADALA learns the same mapping function $F' : S_U \to \hat{S}^z$ as DARLA, also utilizing a $\beta$ variational autoencoder. It then learns a policy $\pi : \hat{S}^z \to A$, but utilizes a soft attention mechanism to ignore irrelevant features. Using this mechanism, it implicitly learns two functions: $\mathcal{A} : \hat{S}^z \to S^a$ that maps from extracted latent state factors to latent state factors relevant to solving the MDP, and $\pi_{\mathcal{A}} : S^a \to A$ that maps from relevant latent state factors to actions.

Soft attention mechanisms learn an importance weight $w_i \in [0, 1]$ for each feature $z_i$ in part of a neural network. These weights are then multiplied element-wise with the corresponding feature. Thus, the attention layer can be defined as follows:

$$f(Z) = W \odot Z,$$

where $W$ is the vector of importance weights $w_i$ and $\odot$ is element-wise multiplication.

$W$ is defined as

$$W = \sigma(h(Z))$$

where $h$ is a differentiable function parameterized by dense layers of a neural network. Thus, the values $w_i \in W$ are constrained between 0 and 1, inclusive. This allows the RL agent to ignore a feature $z_i$ by setting $w_i$ to 0. It also allows the agent to down-weight the importance of a feature by setting $w_i$ to a fractional value or pay full attention by setting $w_i$ to 1.

## 4.3 SADALA FRAMEWORK

The SADALA framework is shown in Figure 1. At inference time, an input state $s \in S$ is fed into a $\beta$ variational autoencoder. The bottleneck representation of the $\beta$-VAE is then used as extracted latent state factors $\hat{S}^z$. Those features are then fed into the attention mechanism, which outputs the weighted features $S^a$, according to the formula $f$ above. Finally, the weighted features are used as the inputs to the deep reinforcement learning algorithm.

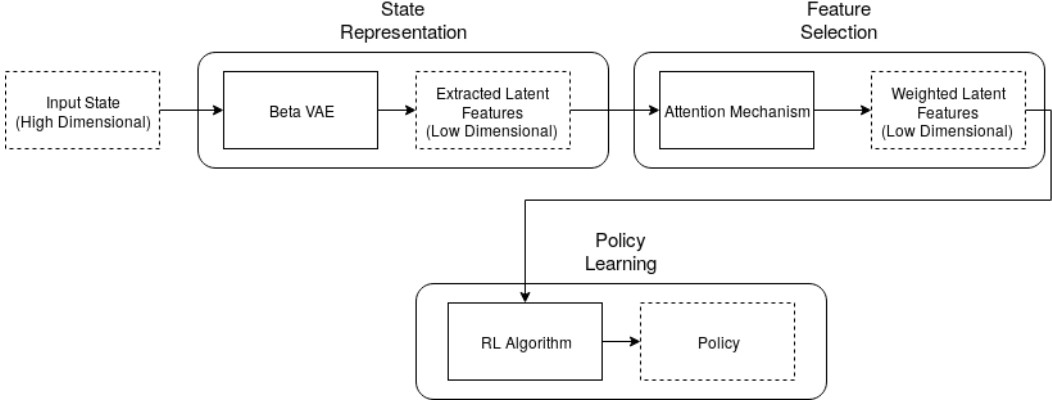

Figure 1: SADALA Framework

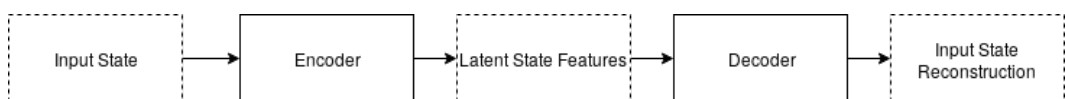

Figure 2: $\beta$ Variational Autoencoder.

Though the feature selection and policy learning stages are represented separately in figure 1, they are contained within the same neural network and optimized jointly.

## 4.4 SADALA TRAINING

Training SADALA requires a few steps. First, the $\beta$ variational autoencoder, shown in figure 2 must be trained. To do so, we use the loss term of a variational autoencoder, modified to fit the $\beta$-VAE (Higgins et al., 2017). The standard VAE loss is

$$\mathcal{L}_{\text{VAE}} = \mathbb{E}_{q(z|x)}[\log p_\theta(x|z)] - D_{KL}(q_\phi(z|x)||p(z)).$$

The $\beta$-VAE loss is

$$\mathcal{L}_{\beta\text{-VAE}} = \mathbb{E}_{q(z|x)}[\log p_\theta(x|z)] - \beta D_{KL}(q_\phi(z|x)||p(z)),$$

where $\beta$ is a weighting term that pressures the encoder $q$ to output a latent space similar to an isotropic unit gaussian. Increasing the value of $\beta$ encourages disentanglement of the latent space while sacrificing minor reconstruction details.

We sample states from the environment, following a random policy. We then train the $\beta$-VAE to reconstruct these images and freeze the network weights. As shown by DARLA, freezing the network weights of the $\beta$-VAE is necessary (Higgins et al., 2017). If the weights are allowed to train jointly with the policy network on the source domain, they will optimize for and overfit to the source domain. When using one source domain, this will cause SADALA to be equivalent to a single-task learner. When using multiple source domains, SADALA would be equivalent to a domain randomization agent.

The feature selection and policy learning stages are then jointly optimized. The SADALA framework can utilize any deep RL algorithm (A3C, REINFORCE, DQN, etc), with minor modification. Specifically, we add a loss term to the loss function $\mathcal{L}_{\text{RL}}$ of the original algorithm. Our new loss is

$$\mathcal{L}_{\text{A-RL}} = \mathcal{L}_{\text{RL}} + |W|,$$

where $W$ is the learned attention weights. This added term is an L1 regularization on the attention weights, which encourages sparsity in feature selection (Kolter & Ng, 2009; Ng, 2004).

We additionally test one other modification to the loss term. We make the assumption that the ordering of the extracted latent state variables $\hat{S}^z$ does not change between subsequent frames within an episode. This, the weight $w_i$ for some latent state factor $z_i$ should not change based on the frame. To enforce this loss on the feature selection stage, we use the following loss

$$\mathcal{L}_{\text{PA-RL}} = \mathcal{L}_{\text{RL}} + |W| + \sum_{j \in \text{len}(z)} \text{Var}[W^j],$$

where $W^j$ is vector of weights corresponding to the $j$th extracted latent state factor vector across the set of training data and Var is the variance. This loss is added to enforce $w_i$ corresponding to the feature $z_i$ to be the same regardless of input. This is similar to learning a pseudo-attention weight vector, rather than an attention weight vector.

See the appendix for additional training details.

## 5 RESULTS AND DISCUSSION

We test the SADALA framework on two transfer learning tasks, using A3C as the deep RL algorithm. The first task is Visual Cartpole. This domain is the same as Cartpole-v1 in OpenAI Gym with two key differences (Brockman et al., 2016). The observed state is now the pixel rendering of the cartpole as well as the velocities of the cart and pole. Thus, the agent must learn to predict the position of the cart and pole from the rendering. Additionally, we modify this domain to include a transfer task. The agent must learn to transfer its knowledge of the game across different color configurations for the cart, pole, and track. Thus, Visual Cartpole is defined as a family of MDPs $M$ where the true state $S^z$ is the positions and velocities of the cart and pole and the observed state $S_U = G_U(S^z)$ for an MDP $U \in M$ is the pixel observations and velocity values. Optimally, the agent should learn to ignore the factors in extracted latent state factors $\hat{S}^z$ that correspond to color, as they do not aid the agent in balancing the pole. This task tests the agent's ability to ignore irrelevant latent state factors.

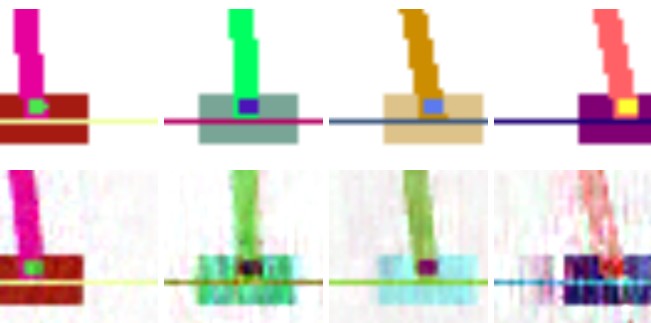

Figure 3: $\beta$-VAE reconstructions of samples from the Visual Cartpole environment. Top row contains original images and the bottom row contains their reconstructions.

The second task is the "Collect Good Objects" task from Deepmind Lab (Beattie et al., 2016). The agent must learn to navigate in first person and pick up "good" objects while avoiding "bad" objects. This task is defined as a family of MDPs $M$ where the true state $S^z$ contains the position of the agent, the position of good and bad objects, the type of good and bad objects, and the color of the walls and floor. In a single MDP $U \in M$, all good objects are hats or all good objects are balloons. Similarly, all bad objects are either cans or cakes. The walls and floor can either take a green and orange colorscheme or a red and blue colorscheme. The agent is trained on hats/cans with the green/orange colorscheme and balloons/cakes with both colorschemes. It is then tested on hats/cans with the red/blue colorscheme. Optimally, the agent should learn to ignore the color of the floor and walls. Additionally, it should use the type of object to determine if it is good or bad. This task tests the agent's ability to ignore distracting latent state factors (the color of the walls and floor) while attending to relevant factors (the positions and types of objects and its own position).

To test the results of the SADALA algorithm, we first test the reconstruction and disentanglement properties of the $\beta$-VAE used in the state representation stage. Note that this stage is identical to that of DARLA  (Higgins et al., 2017). As such, we expect the disentanglement properties to be similar.

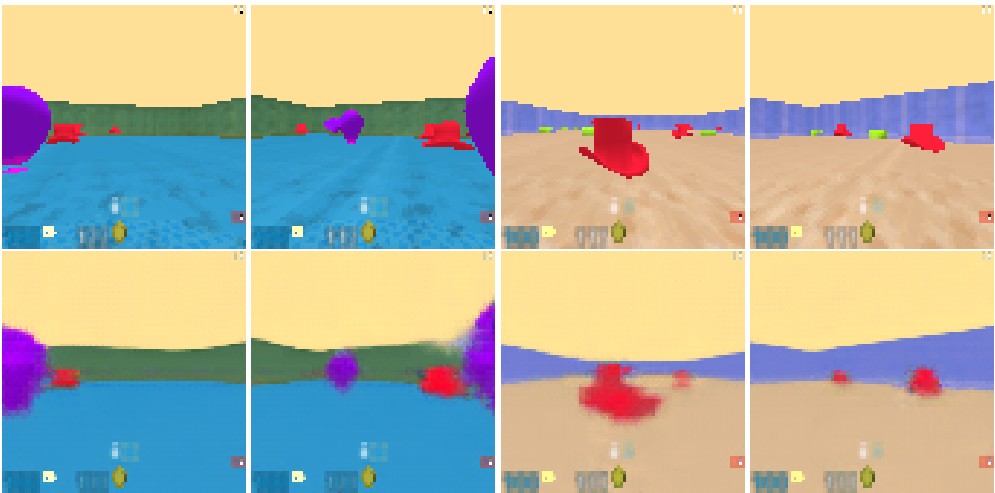

Figure 4: $\beta$-VAE reconstructions of samples from the Deepmind Lab environment. Top row contains original images and bottom row contains their reconstructions.

See figure 3 for reconstructions of the cartpole state. Based on the reconstructions, it is apparent that the $\beta$-VAE has learned to represent cart position and pole angle. Though the angle of the poles is slightly incorrect in the first set of images, the pole is tilted in the correct direction, yielding sufficiently correct extracted latent state factors.

Additionally, the color of the cart, pole, and background is incorrect in the third pair of images. While this demonstrates that the identification and reconstructions of colors is not infallible, the position of the cart and pole remains correct, yielding a set of extracted latent state parameters that is sufficient to solve the MDP.

See figure 5 for a visualization of reconstruction with attention. In the original image, the pole is standing straight and the cart is centered. In the reconstruction, the cart is centered, and the pole is almost upright. However, the reconstruction does not include the colors of the cart or pole. Instead it fills the cart and pole with the mean color of the dataset. This shows that the attention weights are properly learning to ignore color and instead pay attention to the position of the cart and pole.

Figures 6 and 7 gives a comparison of the performance of the algorithms across environments. Note that all of the algorithms are sufficient at solving the source task, with the single-task learner performing slightly better. This is due to the fact that the single-task learner can optimize its con-

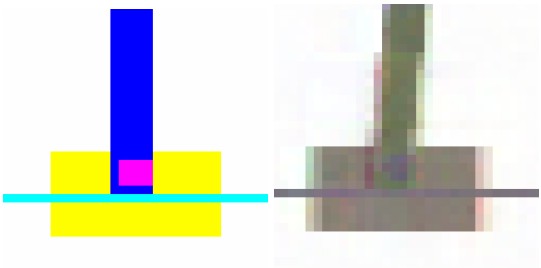

Figure 5: $\beta$-VAE reconstructions from Visual Cartpole. The image on the left is the original image from the simulation and the image on the left is the reconstruction after multiplying with the learned attention weights.

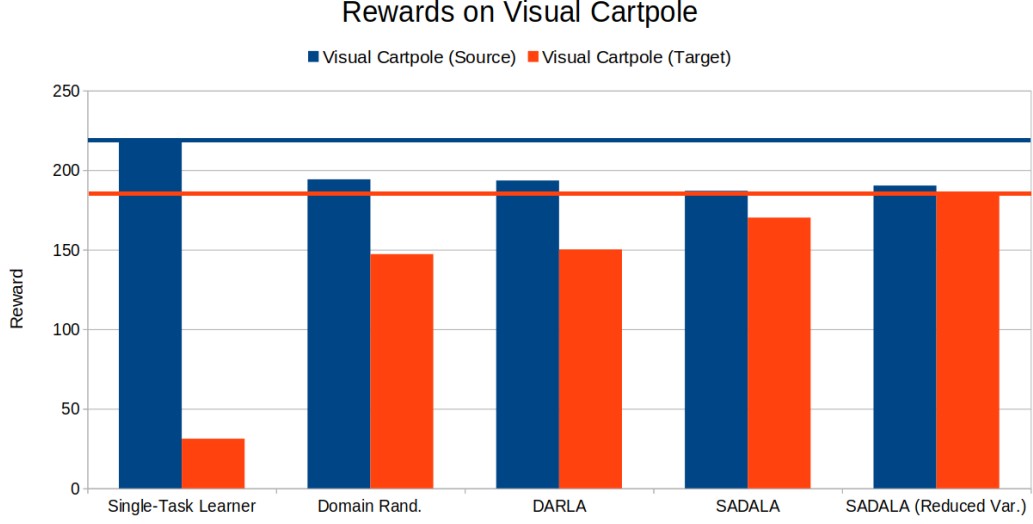

Figure 6: Source and target rewards on Visual Cartpole.

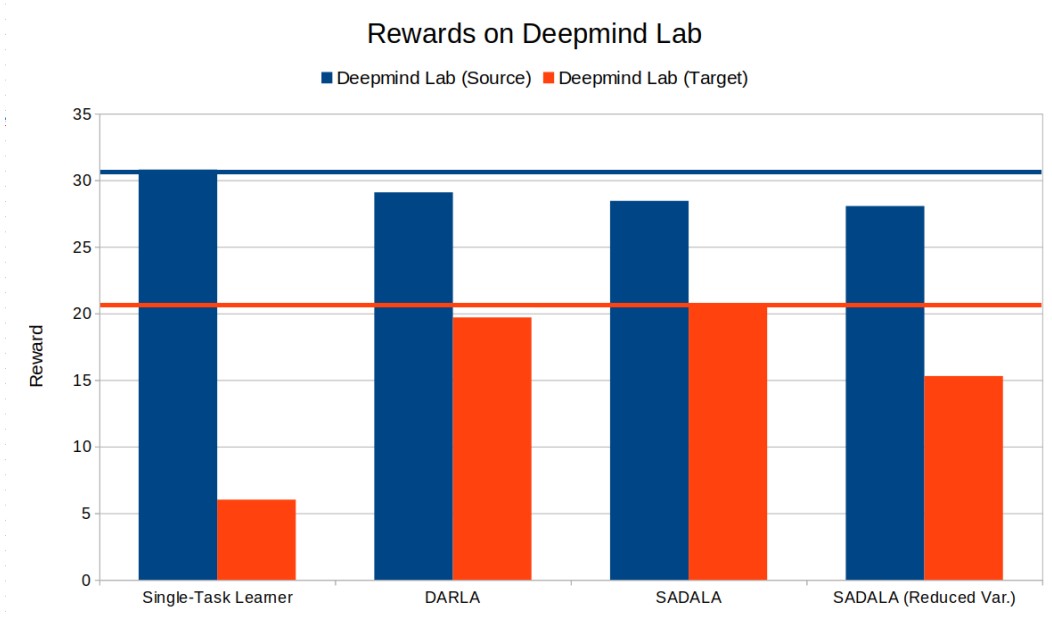

Figure 7: Source and target rewards on Deepmind Lab.

volutional filters to the source domain at the expense of failing to transfer to the target domain. For numerical reward values, see table 1 in the appendix.

The single-task learner achieves better rewards on all source tasks than any of the transfer-specific agents. Domain randomization performs less well because of the complexity of the domain randomization task. Rather than optimizing reward for a single domain, the agent must optimize reward across a large set of domains.

DARLA, SADALA, and SADALA with reduced variance also perform less well on the source task than the baseline agent. This is due to imperfections in the $\beta$-VAE. Though the $\beta$-VAE attempts to reconstruct the input image, it does not do so perfectly, as shown in figure 3. This shows that its

extraction of latent state features is not perfect, leading to potentially confusing states given to the RL agent. Additionally, while the $\beta$-VAE's goal is to learn a disentangled representation, it does not do so perfectly. As such, (partially) entangled latent state factors may further confuse the RL agent.

The single-task learner fails to transfer its policy to the target domain. DARLA transfers some of its knowledge to the target domain. Domain randomization also transfers some knowledge. Finally, SADALA transfers more of its knowledge. The single-task agent has no incentive to learn a factored state representation that enables transfer. Its convolutional filters will directly optimize to maximize reward in the source policy. Thus, if the filters are searching for a hat on a blue floor but tested on a hat on a red floor the convolutional filters will fail to transfer.

DARLA learns a state representation that is mostly disentangled. This allows the RL agent to learn to ignore unimportant features such as cart color and utilize important features such as cart position. The factored state representation forces the RL agent to learn a more robust policy. However, the neural network parameterization of the RL policy must implicitly learn to ignore unimportant factors. Therefore, when presented with unseen information, the RL agent may not properly ignore unimportant factors.

Domain randomization forces the neural network to implicitly learn a state representation sufficient to transfer between tasks. This requires large amounts of training data and is less robust than explicit modeling of latent state factors.

SADALA builds on DARLA by adding an explicit attention mechanism, allowing it to more effectively ignore unimportant features. Due to the use of the sigmoid activation in the attention mechanism, the attention weights $W$ are bounded between $0$ and $1$. In addition to providing a direct weight on the importance of a feature, this bound prevents high variance of attention weights across different inputs.

SADALA with the variance reduction term performs worse than both DARLA and SADALA without variance reduction on the Deepmind lab task but better on the other two. In the scenario where the extracted latent state factors from the $\beta$-VAE are perfectly disentangled, static attention weights should be sufficient to solve the source task and should transfer better to the target task, as in the Visual Cartpole task. However, the $\beta$-VAE does not output perfectly disentangled factors, especially in more complex visual domains such as the Deepmind lab. Thus, the amount of attention paid to each feature from the $\beta$-VAE may differ across tasks, violating the assumption that attention weights should be zero variance.

## 6 CONCLUSION

In this paper we propose SADALA, a three stage method for zero-shot domain transfer. First, SADALA learns a feature extractor that represents input states (images) as disentangled factors. It then filters these latent factors using an attention mechanism to select those most important to solving a source task. Jointly, it learns a policy for the source task that is robust to changes in states, and is able to transfer to related target tasks. We validate the performance of SADALA on both a high-dimensional continuous-control problem (Visual Cartpole) and a 3D naturalistic first-person simulated environments (Deepmind Lab). We show that the attention mechanism introduced is able to differentiate between important and unimportant latent features, enabling robust transfer.

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

# A   APPENDIX

## A.1   TASKS

We test the SADALA framework on two transfer learning tasks, using A3C as the deep RL algorithm. These tasks are described in detail below.

### A.1.1   VISUAL CARTPOLE

As in the original cartpole domain, an agent must learn to keep a pole balanced on a cart upright for as long as possible. The states in the original domain are the position and velocity of the cart and pole, and the states in the modified domain are image renderings of the cart and pole. The agent acts in both domains by applying a force of $+1$ or $-1$ to the cart, and the environment transitions based on a deterministic physics simulation of the cart and pole. The reward is $+1$ at every timestep that the pole has not fallen over, and the episode terminates when the pole falls over or the cart moves too far to the left or right. Similar to OpenAI Gym's Cartpole V1, Visual Cartpole ends after a maximum of $500$ steps, yielding a maximum reward of $500$ per episode (Brockman et al., 2016).

Note that visual cartpole as described above is a Partially Observable Markov Decision Process (POMDP), in which there is a hidden state that is not directly observable. Specifically, it is not possible for an agent to determine the velocity of the cart and pole solely based on one image. In order to simply the visual cartpole task to an MDP, velocity information is included in the state. This enforces the Markov property on the environment, making it better suited to test visual transfer.

When framing this MDP as a transfer learning task, the agent is trained on some number of source domains, where the colors of the cart, pole, axle, and track are varied. Specifically, the color of each element (cart, pole, etc) is sampled randomly from a normal distribution with mean $0.5$ and standard deviation $0.5$ and clipped to the interval $[0, 1]$. Note that each color is represented as a vector of three numbers, each within the interval $[0, 1]$. The colors are randomly sampled at the end of each episode. The agent is then tested on a target domain with held-out colors, which are sufficiently different from those of the source environments. Specifically the colors of the source environments

are constrained to be outside of a hypersphere of radius $0.1$ the test environment. When evaluating source domain performance, the agents trained on multiple source domains are all evaluated on the same domain, randomly sampled from the set of source domains. The single task learner is evaluated on the source domain it is trained on.

### A.1.2 DEEPMIND LAB

The second task is the Collect Good Objects task from Deepmind Lab (Beattie et al., 2016). This environment can be defined as a family of MDPs that share transition function, reward function, and action space. The goal of the agent is to pick up as many "good" objects ($+1$ reward) and avoid picking up "bad" objects ($-1$ reward) as it can within one minute.

These objects can be hats, cans, cakes, and balloons. Additionally the color combinations of the walls and floors can vary between green walls and orange floors or red walls and blue floors. These variations in state space correspond to the different MDPs within the family $N$ of Deepmind Lab MDPs. The $\beta$-VAE is trained on a set of MDPs covering all latent state factor combinations, the RL network and attention mechanism is trained on a subset of $N$ and tested on a held-out MDP.

### A.2 ADDITIONAL RESULTS

Table 1 compares the numerical reward values displayed in figures 6 and 7. The source domains VC (Source) and DM (Source) are listed for completion. It is better to sacrifice some reward in the source domain for additional reward in target domains.

| Task | Single-Task Learner | Domain Rand. | DARLA | SADALA | SADALA (Var. Reduction) |
|------|------|------|------|------|------|
| VC (Source) | 219.53 | 194.2 | 193.48 | 187.0 | 190.3 |
| VC (Target) | 31.2 | 147.2 | 150.136 | 170.16 | **185.87** |
| DM (Source) | 30.8 | — | 29.09 | 28.45 | 28.06 |
| DM (Target) | 6.2 | — | 19.7 | **20.7** | 15.3 |

Table 1: Mean reward of A3C, Domain Randomization, DARLA, and SADALA agents on Visual Cartpole and Deepmind Lab. Higher is better. Each mean reward is taken over a series of 5000 test episodes according to 5 trials.

## B NEURAL NETWORK DETAILS

The neural network for the Visual Cartpole task consists of two parts: the $\beta$-VAE network and the policy network. First, the encoder for the $\beta$-VAE network is composed of three convolutional layers, each with kernel size 3 and stride 1. In order, they have 32, 32, and 64 filters. The output of the last convolutional layer is then frozen and passed through a dense layer with 256 neurons. This is then passed through another dense layer with 64 neurons, which outputs the latent space. The latent space is composed of 32 gaussians, each parameterized by a mean and variance. The decoder is an inverse of the encoder, utilizing deconvolutional layers. The encoder from the $\beta$-VAE is then frozen and used as a feature extractor for the policy network. The policy network takes the encoders outputs and passes them through one dense layer with 64 neurons, which outputs the attention weights. These weights are multiplied elementwise with the encoder's outputs and passed through two dense layers, each with 128 neurons. Finally, the output is passed through a dense layer with two neurons, outputting the policy given the input state. The output is also passed through a dense layer with one neuron, which outputs the value of the input state.

The networks used for the Domain Randomization agent and Single-Task learner have a similar structure. They first pass the input through three convolutional filters, identical in architecture to the encoder. They then pass the outputs of the convolutional filters through two dense layers, each with 128 neurons. Finally, this output is passed through the dense layer with two neurons which outputs the policy, and the dense layer with one neuron, which outputs the value.

The neural networks used for the Deepmind Lab task are slight modifications to the neural networks described above. They have four convolutional layers instead of three, with 32, 32, 64, and 64 filters, respectively. Additionally, the hidden dense layers in the policy network have 256 rather than 128 neurons. As in DARLA, the reconstruction loss of the $\beta$-VAE for Deepmind Lab takes place in the latent space of a denoising autoencoder with 100 latent factors Higgins et al. (2017).

Relu activations are used throughout. All Visual Cartpole networks are trained with Adam with a learning rate of $5e - 4$ and all Deepmind Lab networks are trained Adam with a learning rate of $1e - 4$. The $\beta$-VAE networks are trained on a set of $1,000,000$ images sampled from the observed state space of source policies. The RL networks are trained on $1,000,000$ and $16,000,000$ states, actions, and rewards for the Visual Cartpole and Deepmind Lab tasks, respectively. The domain randomization agent is trained on $3,000,000$ Visual Cartpole states, actions, and rewards.

## C  DOMAIN RANDOMIZATION DETAILS

Domain randomization was implemented by varying parameters me

