# OpenReview forum: "Zero-Shot Policy Transfer with Disentangled Attention"
_ICLR.cc/2020/Conference — Reject_

### Official Review · AnonReviewer2 · 2019-10-21
**Official Blind Review #2**

**Rating:** 1

**Review:**

Pros:
This paper proposed a new method for zero-shot transfer learning under the reinforcement learning setting. The use of attention weights to regularize the latent states was fairly interesting.

Cons:
Limited applicability of the proposed methods
- The paper was restricted in a setting where rewards, actions, and true states were identical between source and target environments, and only the observed states differed due to differing renderers. Working under such a restricted setting was interesting in its own right, but it might also lead to limited applicability of the proposed method in the real-world setting.
- The proposed method focused on solving a very specific problem: learning a dis-entangled latent representation for images. As a result, the potential impact of the proposed methods could be minimal.

Limited technical novelty
- The proposed method, SADALA, was built on top of Higgins et al., 2017 (DARLA). The only difference was an added attention layer to the learning of latent states. As a result, the novelty of the proposed method was very incremental and limited from a technology perspective.
- Even with additional attention layer, the paper could have performed a more thorough study to help the readers understand and appreciate the idea. For example, this paper didn’t discuss the tradeoff between training SADALA over separate stages, versus training it from end to end. For example, why the weights of the pre-trained beta-VAE had to be frozen and used as weights in the state representation stage.

Insufficient experiments
-More thorough discussion of the qualitative results should be helpful to understand whether the attention weights helped the model to focus on the right thing. For example, this paper did study the quality of reconstruction in Figure 3-5 of the proposed method. When comparing Figure 3 and Figure 5, it appeared to me that the reconstructed the angle of the pole was different from the original one. And it seemed like attention weights did successfully ignored the color of the cart and pole, but it ignored the angle of the pole, which should be important to the learning task. Unfortunately, the paper didn't further explain the implication of such misrepresentation.

-Quantitative results
* It would be interesting to all compare the proposed methods against model-agonistic methods like MAML
* It would be useful to include confidence intervals over different tasks.
* It would be useful to compare different methods with different parameter settings
* The authors mentioned “Visual Pendulum tasks” but didn’t include them in the paper


Reproducibility
- It's unclear to me how reproducible the research conducted in this paper was, and it would be useful to open source the code used to conduct the experiments.

**Experience Assessment:**

I have read many papers in this area.

**Review Assessment: Checking Correctness Of Derivations And Theory:**

I assessed the sensibility of the derivations and theory.

**Review Assessment: Checking Correctness Of Experiments:**

I assessed the sensibility of the experiments.

**Review Assessment: Thoroughness In Paper Reading:**

I read the paper at least twice and used my best judgement in assessing the paper.

---

> ### Author Response · Authors · 2019-11-09
> **Responses to your comments & Revision uploaded**
>
> Thank you for your review.
>
> I have uploaded a revision which addresses your comments and also respond to them here.
>
> Limited applicability of the proposed methods:
> - I have now made my focus on this setting clear in the introduction. Other work has shown success in the  problem of transferring to domains with differing dynamics. Thus, I focus on the orthogonal (and yet unsolved) problem of visual domain transfer.
> - In the revised introduction and related work section, I show the importance of learning a dis-entangled representation for images. Not only has much work (now mentioned in related work section) focused on learning a dis-entangled representation, doing so would allow a (visual) Reinforcement Learning agent to learn a sufficient and transferable state representation. The question of how to best learn a state representation (often referred to as state abstraction) is an open problem in Reinforcement Learning
>
> Limited technical novelty:
> - In the related work section, I discuss the problems with DARLA (preserving domain shift, including irrelevant information in the state representation). The attention mechanism directly addresses these problems by ignoring irrelevant information and preserving only information needed to solve the RL task (reducing domain shift).
> - In the SADALA Training section (4.4), I now discuss the tradeoff between training in separate stages vs end-to-end and the importance of weight freezing.
>
> Insufficient experiments:
> - I am currently re-running the experiments to verify their outputs. Shortly, I will upload a revision with the updated results and discussion.
>
> Quantitative results:
> - Model Agnostic Meta Learning (MAML) and related model-agnostic methods do not solve the problem posed in my paper (zero shot transfer). They focus on meta-learning such that a model is able to learn a new task with few samples. This is few-shot learning, not zero-shot transfer.
> - I am not sure which different methods and different parameters refer to. I have compared against DARLA with the parameters used in its paper. Additionally, the parameters for my RL algorithms are fixed across different approaches (DARLA, Domain Randomization, SADALA).
>
> Reproducibility:
> - The original code was open sourced and a link was included in submission to openreview along with the paper. Once the experiments are re-run, I will update the link to the open source code.

---

### Official Review · AnonReviewer3 · 2019-10-23
**Official Blind Review #3**

**Rating:** 1

**Review:**

The paper proposes adding an attention mechanism to the DARLA beta-VAE approach to transfer learning. The beta-VAE, soft attention and policy are trained on appropriate source tasks and evaluated zero-shot on target tasks, using two more difficult continuous control domains with RGB observations. Results indicate some improvements to compared to the immediate relevant baseline which may be statistically significant, but it is not clear whether over 10% in practice.

I cannot at this stage recommend acceptance for the following reasons:
1) The paper augments an existing method with a well understood attention mechanism, so the novelty of the approach is relatively low.
2) The experimental results are interesting, but I don't find them compelling enough to recommend acceptance based on the results alone.  The paper does not solve a major problem with the approach it is based on. In fact, the improvement seems to be smaller when the environment is more complex.
3) Several baselines which are cited in the paper are actually missing in the experiments, so it is hard to determine how important is that roughly 10% improvement compared to SOTA.



**Experience Assessment:**

I have published one or two papers in this area.

**Review Assessment: Checking Correctness Of Derivations And Theory:**

N/A

**Review Assessment: Checking Correctness Of Experiments:**

I assessed the sensibility of the experiments.

**Review Assessment: Thoroughness In Paper Reading:**

I read the paper at least twice and used my best judgement in assessing the paper.

---

> ### Author Response · Authors · 2019-11-09
> **Responses to your comments and Revision uploaded**
>
> Thank you for your review.
>
> I have revised and re-uploaded the paper to address your comments.
>
> 1) To my best knowledge, this paper is the first time in domain adaptation for RL that has explicitly learned a state representation that ignores irrelevant visual features, attention mechanism or not. I have made this clear in the introduction of the re-uploaded paper.
>
> 2)
> - In the (re-uploaded) related work section, I have made the problems with the original approach clear: DARLA preserves domain shift due to the encoding of its state representation. Since its beta-VAE is incentivized to reconstruct the image, it preserves differences between the source and target domains, making transfer difficult. Our approach eliminates this domain shift by learning to attend to only the features relevant to solving the RL task and ignoring all others.
> - I am re-running the experimental results to verify their output (reconstruction and transfer performance). I will upload the results shortly.
>
> 3)
> - I have made clear which of the citations are related work and which are baselines in the (new) related work section. Many of the approaches cited require samples from both the source and target domains and can only transfer to that target domain.
> - The only related works that do not have this problem are DARLA and Domain Randomization, both of which are compared against as baselines.

---

### Official Review · AnonReviewer1 · 2019-10-26
**Official Blind Review #1**

**Rating:** 1

**Review:**

Summarize what the paper claims to do/contribute.
- The paper proposes a new method for zero-shot visual transfer for RL, SADALA. The method first learns a feature extractor with attention (to focus on realted features only) and then learns a policy in the source task and is able to transfer zero-shot int he target domain. The method is evaluated on two tasks: Cartpole-v1 (Gym) and "Collect Good Objects" (Deepmind Lab). It is compared against DARLA for both tasks and against Domain Randomization only for Cartpole.

Clearly state your decision (accept or reject) with one or two key reasons for this choice.
Reject.
- The experiments of the paper were particularly weak.
--More standard visual adaptation techniques like DANNs,ADDA, PixelDA/SimGAN, CycleGAN were not considered.
--The results on domain randomization were not convincing: more details are necessary to determine what the experimental protocol was. One major question: what is the source domain in the case of domain randomization (for Fig. 6) In any case, I find it very hard to believe that simple domain randomization considered here can not fully solve this task for all visual pertrubations considered.
-- In Fig. 5 the reconstruction is not correct.
-- Domain randomization was not tried on the DeepMind Lab example because of compute. However, I'd encourage the authors to try this. Converging will surely not be  linear to the number of perturbations considered as it seems to be implied. Also the OpenAI paper cited as an example where domain randomization took 100 years of simulation required for transfer is a problem of rather different scale: the domain gap there is between simulation and reality for an anthropomorfic robotic hand, and not a simple visual gap where the color of an identical environment are changed.

-Related work discussion was insufficient
-- Related work section is missing and work is not adequately placed in the context of existing literature in the Introduction where some related work is indeed discussed.
-- Related work at the last sentence of the introduction is not discussed correctly. It is implied that all these works are on domain randomization which is not true. Also one work (Chebotar et al) is not relevant as from what I recall there was no visual gap. Finally most of these works deal with much more complex visual gaps so sample complexity is hard to be compared.


**Experience Assessment:**

I have published in this field for several years.

**Review Assessment: Checking Correctness Of Derivations And Theory:**

I did not assess the derivations or theory.

**Review Assessment: Checking Correctness Of Experiments:**

I carefully checked the experiments.

**Review Assessment: Thoroughness In Paper Reading:**

I made a quick assessment of this paper.

---

> ### Author Response · Authors · 2019-11-09
> **Related work discussion has been added**
>
> Thank you for your review.
>
> - I have re-uploaded the paper and addressed your concerns. I have added a related work section, placing my work in the context of exiting literature, adding other relevant work.
> -- DANNs, ADDA, PixelIDA/SimGAN, and CycleGAN are discussed in this section. They focus on visual domain adaptation where samples from the target domain are present during training time and transfer is only to that target domain.
> -- As mentioned in this section, I am solving a different problem, where samples from the target domain are not present during training time, similar to DARLA. Thus, it is good to discuss these approaches, but not necessary to empirically compare against them.
>
> - I have added additional experimental details in the appendix. Specifically, methods requiring multiple source domains to train are evaluated (in fig 6) on one domain randomly sampled from the set of source domains. This sample is the same for the evaluation of all algorithms.
>
> - I am currently running domain randomization for deepmind lab and will upload results shortly.
> - I am re-running experiments (particularly for the reconstruction of figure 5) to verify their output and will upload results shortly.

---

### Decision · Program_Chairs · 2019-12-19

**Decision:**

Reject

**Comment:**

This paper proposes a new method for zero-shot policy transfer in RL. The authors propose learning the policy over a disentangled representation that is augmented with attention. Hence, the paper is a simple modification of an existing approach (DARLA). The reviewers agreed that the novelty of the proposed approach and the experimental evaluation are limited. For this reason I recommend rejection.